



# University of Warsaw Lagrangian Cloud Model (UWLCM) 2.0: Adaptation of a mixed Eulerian-Lagrangian numerical model for heterogeneous computing clusters

Piotr Dziekan and Piotr Zmijewski

Institute of Geophysics, Faculty of Physics, University of Warsaw, Poland

**Correspondence:** Piotr Dziekan (pdziekan@fuw.edu.pl)

**Abstract.**

A numerical cloud model with Lagrangian particles coupled to an Eulerian flow is adapted for distributed memory systems. Eulerian and Lagrangian calculations can be done in parallell on CPUs and GPUs, respectively. Scaling efficiency and the amount of parallelization of CPU and GPU calculations both exceed 50% for up to 40 nodes. A sophisticated Lagrangian
microphysics model slows down simulation by only 50% compared to a simplistic bulk microphysics model, thanks to the use of GPUs. Overhead of communications between cluster nodes is mostly related to the pressure solver. Presented method of adaptation for computing clusters can be used in any numerical model with Lagrangian particles coupled to an Eulerian fluid flow.

## 1 Introduction

As the Moore's law has come to an end, new modeling techniques are required to match the demand for more precise numerical simulations of physical processes (Bauer et al., 2021). We present an implementation of the cloud model UWLCM for distributed memory systems that uses some of the modeling techniques reviewed by Bauer et al. (2021): use of heterogeneous clusters (with parallel computations on CPU and GPU), mixed-precision computations, semi-implicit solvers, different time steps for different processes, portability to different hardware. Although we discuss a numerical cloud model, conclusions and
the techniques used can be applied to modeling of other processes in which Lagrangian particles are coupled to an Eulerian field, such as the particle-in-cell method used in plasma physics (Hockney and Eastwood, 1988).

In numerical models of the atmosphere, clouds are represented using various approximations depending on resolution of the model. In large scale models, like global climate and weather models, clouds are described with a simplistic process, what is known as cloud parameterization. Cloud parameterizations are developed based on observations, theoretical insights and on
fine scale numerical modeling. Therefore correct fine scale modeling is important for better understanding of Earth's climate and for better weather prediction. Highest resolution numerical modeling is known as direct numerical simulation (DNS). In DNS, even the smallest turbulent eddies are resolved, what requires spatial resolution in the millimeter range. Largest current DNS simulations model a volume of the order of several cubic meters, not enough to capture many important cloud scale processes. Whole clouds and cloud fields can be modeled with the the Large Eddy Simulations (LES) technique. In LES, small

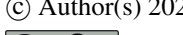



scale eddies are parameterised, so that only large eddies, typically of the order of tens of meters, are resolved. Thanks to this, it is feasible to model a domain spanning tens of kilometers.

DNS and LES models of clouds need to resolve air flow, what is referred to as cloud dynamics, and evolution of cloud droplets, what is known as cloud microphysics. UWLCM is a tool for LES of clouds with a focus on detailed modeling of cloud microphysics. Dynamics are represented in an Eulerian manner. Cloud microphysics are modeled in a Lagrangian particle-

based manner based on the Super-Droplet Method (SDM) (Shima et al., 2009). Lagrangian particle-based cloud microphysics models have gained in popularity in the last decade (Shima et al., 2009; Andrejczuk et al., 2010; Riechelmann et al., 2012). These are very detailed models applicable both to DNS and LES. Their level of detail and computational cost are comparable to the more traditional Eulerian bin models, but Lagrangian methods have several advantages over bin methods (Grabowski et al., 2019). Simpler, Eulerian bulk microphysics schemes are also available in UWLCM.

We start with a brief presentation of the model, with particular attention given to the way the model was adapted to distributed memory systems. This was done using a mixed OpenMP + Message Passing Interface (MPI) approach. Next, model performance is tested on standalone servers followed by tests on a computing cluster. Main goals of these tests are to determine simulation parameters that give optimal use of computing hardware and check model scaling efficiency. Other discussed topics are the GPU vs CPU speedup and performance of different MPI implementations.

## 2  Model description

Full description of UWLCM can be found in Dziekan et al. (2019). Here, we briefly present key features. Cloud dynamics are modeled using an Eulerian approach. Eulerian variables are the flow velocity, potential temperature and water vapor content. Equations governing time evolution of these variables are based on the Lipps-Hemler anelastic approximation (Lipps and Hemler, 1982), which is used to filter acoustic waves. These equations are solved using a finite difference method. Spatial

discretization of the Eulerian variables is done using the staggered Arakawa-C grid (Arakawa and Lamb, 1977). Integration of equations that govern transport of Eulerian variables is done with the multidimensional positive-definite advection transport algorithm (MPDATA) (Smolarkiewicz, 2006). Forcings are applied explicitly with the exception of buoyancy and pressure gradient, which are applied implicitly. Pressure perturbation is solved using the generalized conjugate residual solver (Smolarkiewicz and Margolin, 2000). Diffusion of Eulerian fields caused by subgrid-scale (SGS) turbulence can be modeled with a

Smagorinsky-type model (Smagorinsky, 1963) or with the implicit LES approach (Grinstein et al., 2007).

Cloud microphysics can be modeled with a single- or double-moment bulk scheme, or with a Lagrangian particle-based model. Depending on the microphysics model, simulations are named UWLCM-B1M (single-moment bulk scheme), UWLCM-B2M (double-moment bulk scheme) or UWLCM-SDM (super-droplet method). Details of microphysics models can be found in Arabas et al. (2015). In both bulk schemes, cloud water and rain water mixing ratios are prognostic Eulerian variables. In the

double-moment scheme, cloud droplet and rain drop concentrations are also prognostic Eulerian variables. In the Lagrangian particle-based scheme, all hydrometeors are modeled in a Lagrangian manner. The scheme is based on the super-droplet method (SDM) (Shima et al., 2009). In particular, it employs the the all-or-nothing coalescence algorithm (Schwenkel et al., 2018).





In SDM, a relatively small number of computational particles, called super-droplets (SD), represent the vast population of all droplets. Equations that govern behaviour of SD are very similar to the well-known equations that govern behaviour of real droplets. The condensation equation includes the Maxwell-Mason approximation and the $\kappa$-Köhler parameterization of water activity (Petters and Kreidenweis, 2007). SD follow the resolved, large-scale flow and sediment at all times with the terminal velocity. Velocity of SD associated with SGS eddies can be modeled as an Ornsetin-Uhlenbeck process (Grabowski and Abade, 2017). Collision-coalescence of SD is treated as a stochastic process in which probability of collision is proportional to the collision kernel. All particles, including humidified aerosols, are modeled in the same way. Therefore, particle activation is resolved explicitly, what often requires short time steps for solving the condensation equation. Short time steps are sometimes also required when solving collision-coalescence. To permit time steps for condensation and collision-coalescence shorter than for other processes, two separate substepping algorithms, one for condensation and one for collision-coalescence, are implemented.

Equations for the Eulerian variables, including cloud and rain water in bulk microphysics, are solved by a CPU. Lagrangian microphysics can be modeled either on a CPU or on a GPU. In the latter case, information about super-droplets is stored in the GPU memory and GPU calculations can be done parallelly to the CPU calculations of Eulerian variables. Compared to UWLCM 1.0 described in Dziekan et al. (2019), order of operations has been changed to allow for GPU calculations to continue in parallel to the CPU calculations of the Eulerian SGS model. Details are given in appendix C.

All CPU computations are done in double precision. Most of the GPU computations are done in single precision. The only exception are high order polynominals, e.g. in the equation for terminal velocity of droplets, that are done in double precision.

So far, UWLCM has been used to model stratocumuli (Dziekan et al., 2019, 2021b), cumuli (Grabowski et al., 2019; Dziekan et al., 2021b) and raising thermals (Grabowski et al., 2018).

## 3 Adaptation to distributed memory systems

Strategy of adapting UWLCM to distributed memory systems was developed with a focus on UWLCM-SDM simulations with Lagrangian microphysics computed by GPUs. Therefore, this most complicated case is discussed first. Simpler cases with microphysics calculated by CPUs will be discussed afterwards.

The difficulty in designing a distributed memory implementation of code in which CPUs and GPUs simultaneously conduct different tasks is in obtaining a balanced workload distribution between different processing units. This is because GPUs have higher throughput than CPUs, but the GPU memory is rather low, what puts an upper limit on the GPU workload. Taking this into account, we chose to use a domain decomposition approach that is visualized in fig. 1. The modeled domain is divided into equal slices along the horizontal axis $x$. Computations in each slice are done by a single MPI task, which can control multiple GPUs and CPU threads. Cloud microphysics within the slice are calculated on GPUs, with super-droplets residing in GPU memory. Eulerian fields in the slice reside in system memory and their evolution is calculated by CPU threads. Since the CPU and GPU data attributed to a task are colocated in the modeled space, all CPU-to-GPU and GPU-to-CPU communications happen via PCI-Express and do not require inter-node data transfer. The only inter-node communications are CPU-to-CPU and





GPU-to-GPU. If an MPI task controls more than one GPU, computations within the subdomain of that task are divided among the GPUs also using domain decomposition along the $x$ axis. An MPI task will typically control more than one CPU thread, because usually cluster nodes have more CPU cores than GPUs. Computations are divided between CPU threads of a task using domain decomposition of the task's subdomain, but along the $y$ axis. Thanks to that, the maximum number of GPUs that can be used in a simulation is equal to the number of cells in the $x$ direction even though there are multiple CPU threads per GPU. MPI communications are done using two communicators, one for the Eulerian data and one for the Lagrangian data. Transfers of the Eulerian data are handled simultaneously by two threads, one for each boundary that is perpendicular to the $x$ axis. This requires that the MPI implementation supports the MPI_THREAD_MULTIPLE thread level. Transfers of the Lagrangian data are handled by the thread that controls the GPU that is on the edge of the task's subdomain. Collective MPI communication is done only on the Eulerian variables and most of it is associated with solving the pressure problem.

It is possible to run simulations with microphysics, either Lagrangian particle-based or bulk, computed by CPUs. In the case of bulk microphysics, microphysical properties are represented by Eulerian fields that are divided between tasks and threads in the same manner as described in the previous paragraph, i.e. like the Eulerian fields in UWLCM-SDM. In UWLCM-SDM with microphysics computed by CPUs, all microphysical calculations in the subdomain belonging to a given MPI task are divided amongst task's threads by the NVIDIA Thrust library (Bell and Hoberock, 2012).

File output is done in parallel by all MPI tasks using the parllel HDF5 C++ library (The HDF Group).

## 4 Performance tests

### 4.1 Simulation setup

Model performance is tested in simulations of a raising moist thermal (Grabowski et al., 2018). In this setup, an initial spherical perturbation is introduced to a neutrally stable atmosphere. Within the perturbation, water vapour content is increased to obtain RH=100%. With time, the perturbation is lifted by buoyancy and water vapor condenses within it. We chose this setup, because it has significant differences in buoyancy and cloud formation already at the start of a simulation. This puts pressure solver and microphysics model to test without need of a spinup period.

Subgrid-scale diffusion of Eulerian fields is modeled with the Smagorinsky scheme. SGS motion of hydrometeors is modeled with a scheme described in (Grabowski & Abade 17). Model time step length is $0.5\,\mathrm{s}$. Substepping is done to achieve a time step of $0.1\,\mathrm{s}$ for condensation and coalescence. These are values typically used when modeling clouds with UWLCM. No output of model data is done.

### 4.2 Computers used

Performance tests were ran on three systems: *Rysy*, *a02* and *Prometheus*. Hardware and software of these systems is given in table 1 and table 2, respectively. *Rysy* and *a02* were used only in the single-node tests, while *Prometheus* was used both in single- and multi-node tests. *Prometheus* has 72 GPU nodes connected with Infiniband. We chose to use the MVAPICH2





2.3.1 MPI implementation on *Prometheus*, because it supports the MPI_THREAD_MULTIPLE thread level, is CUDA-aware and is free to use. Other implementation that meets these criteria is OpenMPI, but it was found to give lower performance in scaling tests of *libmpdata++* ( appendix B). NVIDIA GPUDirect and GDRCopy technologies were not used by the MPI

implementation, because they are not supported by MVAPICH2 for the type of interconnect used on *Prometheus*. MVAPICH2 does not allow more than one GPU per task. Therefore multi-node tests were done for 2 tasks per node, each task controlling 1 GPU and 12 threads.

## 4.3  Single-node performance

In this section we present tests of computational performance of UWLCM-SDM ran on a single server. The goal is to determine

how parallellization of CPU and GPU computations can be maximized. We also estimate the speedup achieved thanks to the use of GPUs. No MPI communications are done in these tests.

Size of the computational grid is 128x128x128. In the super-droplet method, quality of microphysics solution depends on the number of super-droplets used. We denote the initial number of super-droplets per cell by $N_{\mathrm{SD}}$. We perform test for different values of $N_{\mathrm{SD}}$. The maximum possible value of $N_{\mathrm{SD}}$ depends on the amount of GPU memory available. When ran on a single

server, all Eulerian fields are stored in shared memory.

Timings presented in figures where obtained with a built-in timing functionality of UWLCM. This timing can be enabled at compile-time by setting UWLCM_TIMING CMake variable. Additional analyses where done with the Arm MAP and Intel vTune performance profilers. The average wall time it takes to do one model time step is plotted in fig. 2. This is separated into time spent on CPU, GPU and parallel CPU&GPU calculations. Complexity of CPU computations depends on the grid size

and, ideally, does not depend on $N_{\mathrm{SD}}$. In reality, we see that CPU computations time slightly increases with $N_{\mathrm{SD}}$. Complexity of GPU computations increases linearly with $N_{\mathrm{SD}}$ and it is seen that GPU time in fact increases linearly with $N_{\mathrm{SD}}$. For low $N_{\mathrm{SD}}$ majority of time is spent on CPU computations. As $N_{\mathrm{SD}}$ increases, total time does not increase much, but the amount of parallel CPU and GPU computations increases until pure GPU computations start to dominate and parallellization starts to decrease again. This is not observed on the *Prometheus* server, because maximum $N_{\mathrm{SD}}$ on it is only 25 due to the limited GPU

memory. Maximum CPU and GPU parallellization is achieved for $N_{\mathrm{SD}} = 32$ and $N_{\mathrm{SD}} = 64$ on *a02* and *Rysy*, respectively. This difference comes from differences in relative CPU to GPU computational power between these machines. In LES, $N_{\mathrm{SD}}$ is usually between 30 and 100. The test shows that high parallellization of CPU and GPU computations, up to 80 %, can be obtained in typical cloud simulations.

In UWLCM-SDM, microphysical computations can also be done by the CPU. From the user perspective, all that needs to be

done is to specify *--backend=OpenMP*. We do so to test how much speedup is achieved thanks to the use of GPUs. We define the GPU vs CPU speedup as the number of CPUs needed per single GPU to obtain the same wall time, assuming that wall time scales perfectly with added CPUs: $(N_{\mathrm{CPU}}/N_{\mathrm{GPU}})(t_{\mathrm{CPU}}/t_{\mathrm{GPU}} - 1)$, where $N_{\mathrm{CPU}}$ ($N_{\mathrm{GPU}}$) is the number of CPUs (GPUs) on the server and $t_{\mathrm{CPU}}$ ($t_{\mathrm{GPU}}$) is the average wall time per time step when using only CPUs (CPUs plus GPUs) for computations. The GPU vs CPU speedup for different machines and different values of $N_{\mathrm{SD}}$ is plotted in fig. 3. It is seen that between 10 to





60 CPUs would be needed to replace a single GPU. We conclude that GPUs provide substantial benefits in equipment cost and power usage.

### 4.4    Multi-node performance

Computational performance of UWLCM-SDM, UWLCM-B1M and UWLCM-B2M on distributed memory systems is discussed in this section. We consider four scenarios in which UWLCM is ran on a distributed memory system for different

reasons:

- *strong scaling* - More nodes are used in order to decrease the time it takes to complete the simulation.

- *SD scaling* - More nodes are used to increase the total GPU memory, allowing for more SD to be modeled, while grid size remains the same. This results in weak scaling of GPU workload and strong scaling of CPU workload. This test is applicable only to UWLCM-SDM.

- *2D grid scaling* - As more nodes are used, the number of grid cells in the horizontal directions is increased, while the number of cells in the vertical is constant. In UWLCM-SDM, number of SD per cell is constant. Therefore, as more cells are added, the total number of SD in the domain increases. This results in weak scaling of both CPU and GPU workloads. This test represents two use cases: domain size increase and horizontal resolution refinement. Typically in cloud modeling, domain size is increased only in the horizontal because clouds form only up to certain altitude.

- *3D grid scaling* - similar to *2D grid scaling*, but more cells are used in each dimension. This would typically be used to increase resolution of a simulation.

Details of the simulation setup for each case are given in table 3. In each case, the maximum number of super-droplets that fit the GPU memory is used in UWLCM-SDM. The only exception is the *strong scaling* test in which, as more nodes are added, the number of SD per GPU decreases. Note how *SD scaling* is similar to *strong scaling*, but with more SDs added as

more GPUs are added. Also note that the *2D grid scaling* and *3D grid scaling* tests are similar, but with differences in sizes of distributed memory data transfers.

UWLCM-SDM simulation time versus number of nodes used is plotted in fig. 4. First, we discuss the *strong scaling* and *SD scaling* tests. When ran on a single node, most of the time is spent on CPU-only computations, while parallel CPU&GPU computations take only about 5% of the total simulation time and the GPU-only computation time is negligible. This indicates

that the ratio of workload to computational power is much higher on CPU than on GPU. GPU workload can only be increased by using more nodes, because of the limited GPU memory. In *strong scaling* test, using more nodes decreases workload for both CPU and GPU, what results in shorter simulation times. Speedup obtained by adding more nodes decreases as more nodes are used, what is indicated by the decreasing scaling efficiency. The decrease in scaling efficiency is mostly caused by poor scaling of collective MPI communications done by the pressure solver. In *SD scaling* tests, the CPU workload scales

the same as in the *strong scaling* tests, but the workload per GPU remains constant. CPU computation time is practically the same as in *strong scaling*. The total time of GPU computations is approximately constant, what shows that there is little cost





of MPI communications between GPUs. Scaling efficiency and the amount of CPU and GPU parallelization is higher than in the *strong scaling* scenario. *SD scaling* is more preferable than *strong scaling*, since the former gives better resolution of cloud microphysics at a very similar total computation time.

2D and *3D grid scaling* are weak scaling tests which differ in the way the size of CPU MPI communications scales. As in the *strong* and *SD scaling* tests, there is no visible cost of MPI communications between GPUs. The total CPU-only computation time increases as more nodes are added, because of the cost of MPI data transfers between system memory of different nodes. Majority of this cost is associated with collective communications necessary to solve the pressure perturbation equation. Altogether, a decrease in scaling efficiency and the amount of CPU and GPU parallellization is observed as more nodes are used. For 36 nodes, that is 72 GPUs and 864 CPU threads, scaling efficiency is above 60% and CPU and GPU parallelization is around 50%.

Comparison of wall time scaling in UWLCM-B1M, UWLCM-B2M and UWLCM-SDM is shown in fig. 5. UWLCM-B1M and UWLCM-B2M use simple microphysics schemes that are computed by the CPU. UWLCM-B2M, which has four Eulerian prognostic variables for microphysics, is more complex than UWLCM-B1M, which has two. Regardless of this, wall time is very similar for UWLCM-B1M and UWLCM-B2M. Wall time of UWLCM-SDM, which uses much more complex microphysics scheme, is higher by a factor that depends on the number of SD. In UWLCM-SDM *2D grid scaling* and *3D grid scaling* tests there are 100 SD per cell, what is a typical value used in LES. Then, UWLCM-SDM simulations take approximately 50% longer than UWLCM-B1M or UWLCM-B2M simulations. The *strong scaling* UWLCM-SDM test uses 3 SD per cell and the wall time is very similar to that of UWLCM-B1M and UWLCM-B2M. Scaling of wall time with the number of nodes is similar for all types of microphysics. This is because the part of the model that scales the worst is the pressure solver, which requires collective MPI communications.

## 5 Summary

A numerical model with Lagrangian particles embedded in an Eulerian fluid flow has been adapted to clusters equipped with GPU accelerators. The Eulerian and Lagrangian computations are parallelly done on CPUs and GPUs, respectively. Computations are distributed among processes using static domain decomposition.

Performance tests for computers with shared system memory were done on three different servers, each equipped with multiple GPUs. Maximum amount of parallelization, up to more than 80%, is obtained for an optimal ratio of CPU and GPU workloads, which depends on the relative computational power of CPU and GPU. The optimal ratio of workloads is obtained for simulation parameters typically used in literature. To complete the simulation in the same amount of time, but without GPU accelerators, each GPU would need to be replaced by even up to 60 CPUs. That would require a distributed memory system, which would incur additional overhead. This shows that GPU accelerators enable us to run a useful scientific simulation on a single server, what would otherwise require a more expensive and more energy-consuming CPU cluster.

Computational performance of the model on a distributed memory system was tested on the *Prometheus* cluster. Cost of communication between nodes slows down computations related to the Eulerian part of the model by a much higher factor



than computations related to the Lagrangian part of the model. This is because solving the pressure perturbation, what is done by the Eulerian component, requires collective communications, while the Lagrangian component requires peer-to-peer communications only. In most cases, scaling efficiency and the amount of CPU and GPU parallelization exceed 50% for up to 40 nodes, with 960 CPU threads and 80 GPUs. A simulation with 20 million grid cells and 2 billion particles can be done in real time. Thanks to the use of GPUs, a sophisticated microphysics model does not slow down simulation much. Simulations

with Lagrangian microphysics computed by GPUs are only 50% slower than simulations with bulk microphysics computed by CPUs.

Our approach of using CPUs for Eulerian calculations and GPUs for Lagrangian calculations gives good amount of parallelization of CPU and GPU calculations and good scaling on computing clusters with several dozen nodes. The same approach can be used in other numerical models with Lagrangian particles embedded in an Eulerian flow.

*Code and data availability.* Source code of UWLCM, libmpdata++ and libcloudph++ is available at https://github.com/igfuw. In the study, the following code versions were used: UWLCM v2.0 (Dziekan and Waruszewski, 2021), libmpdata++ v2.0-beta (Arabas et al., 2021) and libcloudph++ v3.0 (Dziekan et al., 2021a). Dataset, run scripts and plotting scripts are available at Dziekan and Zmijewski (2021).

## Appendix A: Software

UWLCM is written in C++14. It makes extensive use of two C++ libraries that are also developed at the Faculty of Physics

of the Universitry of Warsaw: libmpdata++ (Jaruga et al., 2015; Waruszewski et al., 2018) and libcloudph++ (Arabas et al., 2015; Jaruga and Pawlowska, 2018). Libmpdata++ is a collection of solvers for generalised transport equations that use the Multidimensional Positive Definite Advection Transport Algorithm (MPDATA) algorithm. Libcloudph++ is a collection of cloud microphysics schemes.

In libcloudph++, the particle-based microphysics algorithm is implemented using the NVIDIA Thrust library. Thanks to

that, the code can be ran on GPUs as well as on CPUs. It is possible to use multiple GPUs on a single machine, without MPI. Then, each GPU is controlled by a separate thread and communications between GPUs are done with asynchronous *cudaMemcpy*. Libmpdata++ uses multidimensional array containers from the blitz++ library (Veldhuizen, 1995). Threading can be done either with OpenMP, Boost.Thread or std::thread. In UWLCM we use the OpenMP threading as it was found to be the most efficient. Output in UWLCM is done using the HDF5 output interface that is a part of libmpdata++. It is based on

the thread safe version of the C++ HDF5 library. UWLCM, libcloudph++ and libmpdata++ make use of various components of the Boost C++ library (Koranne, 2011). In order to have parallel CPU and GPU computations in UWLCM, functions from libmpdata++ and from libcloudph++ are launched using std::async. UWLCM, libcloudph++ and libmpdata++ are open source software distributed via the Github repository https://github.com/igfuw/. They have test suits that are automatically ran on Github Actions. To facilitate deployment, a Singularity container with all needed dependencies is included in UWLCM

(https://cloud.sylabs.io/library/pdziekan/default/uwlcm).

Libcloudph++ and libmpdata++ have been adapted to work on distributed memory systems. This has been implemented using the C interface of MPI in libcloudph++ and the Boost.MPI library in libmpdata++. Tests of scalability of libmpdata++ are presented in appendix B.

## Appendix B: Scalability of libmpdata++

UWLCM uses the libmpdata++ library for solving the equations that govern time evolution of Eulerian variables. The library had to be adapted for work on distributed memory systems. The domain decomposition strategy is as in fig. 1, but without GPUs. Here, we present strong scaling tests of standalone libmpdata++. The tests are done using a dry planetary boundary layer setup, which is a part of the libmpdata++ test suite. Grid size is 432x432x51. Tests were done on the *Prometheus* cluster. Note that all libmpdata++ calculations are done on CPUs. Two implementations of MPI are tested, OpenMPI v4.1.0 and

MVAPICH2 v2.3.1. Note that *Prometheus* has 2 GPU per node, but MVAPICH2 does not support more than 1 GPU per task, so 2 tasks per node would need to be ran in UWLCM-SDM. OpenMPI does not have this limitation. For this reason in the libmpdata++ scalability tests we consider two scenarios, one with two processes per node and the other with one process per node. In the case with two processes per node, each process controls half of the available threads. Test results are shown in fig. A1. In general, better performance is seen with MVAPICH2 than with OpenMPI. Running two tasks per node improves

perfromance in MVAPICH2, but decreases performance in OpenMPI. In the best case, scaling efficiency exceeds 80% for up to 500 threads.

## Appendix C: UWLCM: order of operations

Order of operations within a time step has changed since Dziekan et al. (2019). The changes were made to increase parallelization of CPU and GPU computations. This is done by continuing GPU computations during SGS model computations done by

the CPU. An updated UML sequence diagram is shown in fig. A2.

*Author contributions.* PD developed the model, planned the described work, conducted simulations and wrote the manuscript. PZ took part in conducting simulations and in writing the manuscript.

*Competing interests.* No competing interests are present

*Acknowledgements.* Initial work on implementation of MPI in libmpdata++ was done by Sylwester Arabas. We thank SA for consulting

contents of the paper. This research was supported by the Polish National Science Center grant no 2018/31/D/ST10/01577, by the PLGrid





Infrastructure, by the Interdisciplinary Centre for Mathematical and Computational Modelling of the University of Warsaw and by the HPC systems of the National Center for Atmospheric Research, Boulder, Co, USA.

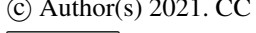



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



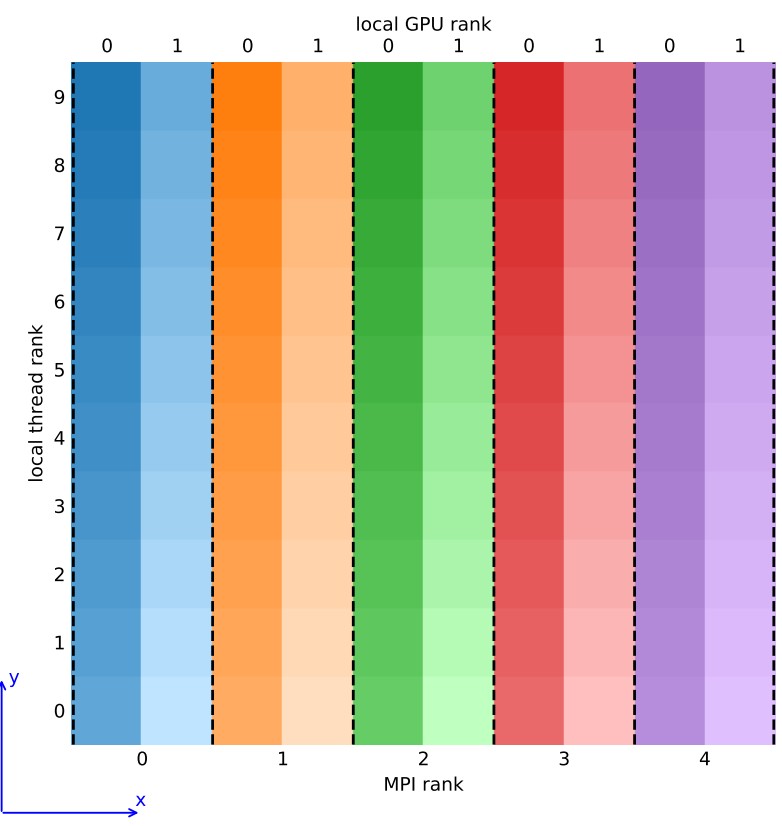

**Figure 1.** Visualization of the domain decomposition approach of UWLCM. Top-down view on a grid with 10 cells in each horizontal direction. Computations are divided among 5 MPI tasks, each controlling 2 GPUs and 10 CPU threads. Local thread/GPU rank is the rank within respective MPI task. Dashed lines represent boundaries over which communications need to be done using MPI assuming periodic horizontal boundary conditions.



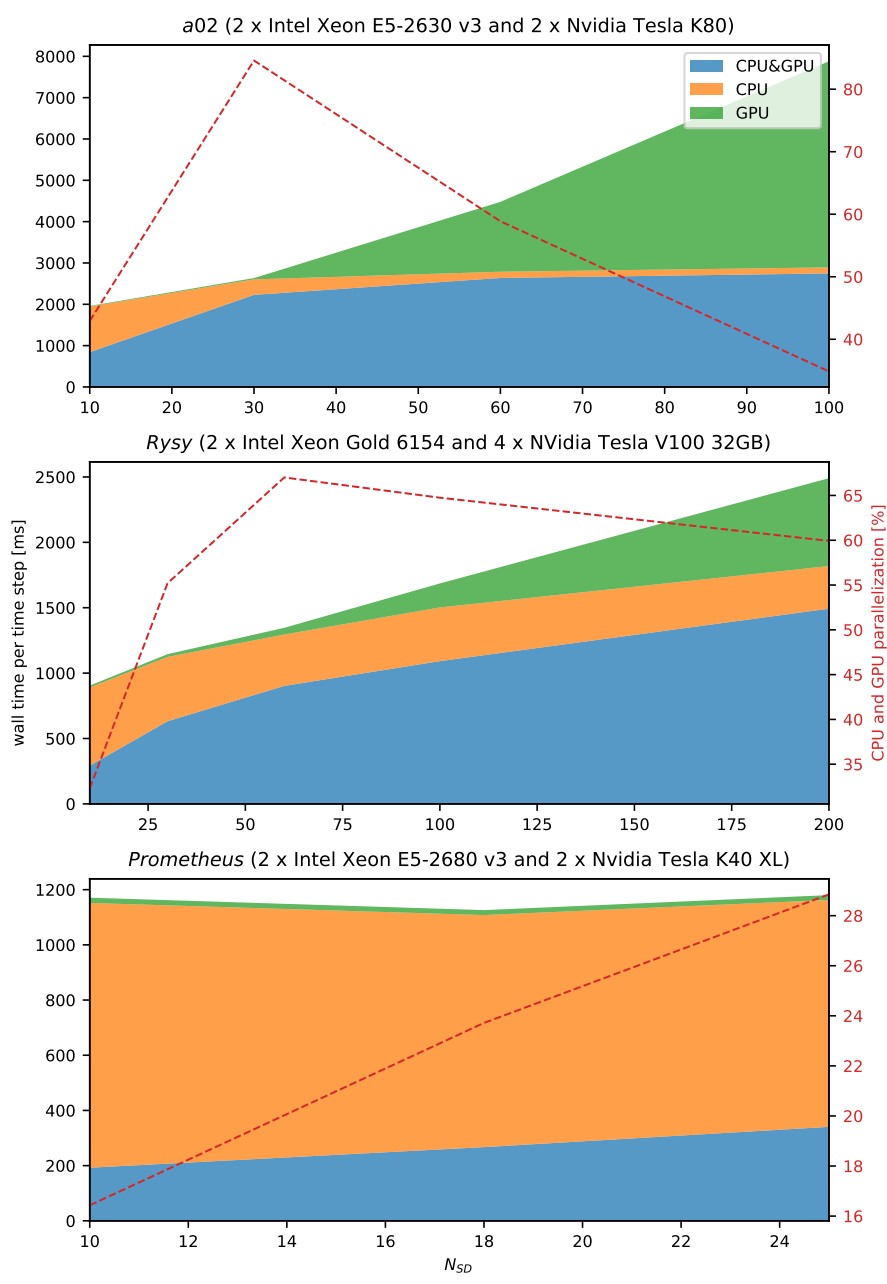

**Figure 2.** Single-node (no MPI) UWLCM-SDM performance for different hardware. Wall time per model time step averaged over 100 time steps. Results of LES of a raising thermal done on three different servers for varying number of super-droplets, $N_{SD}$. Execution time is divided into time spent on CPU computations, GPU computations and parallel CPU and GPU computations. The dashed red line is the percentage of time spent on parallel CPU and GPU computations.





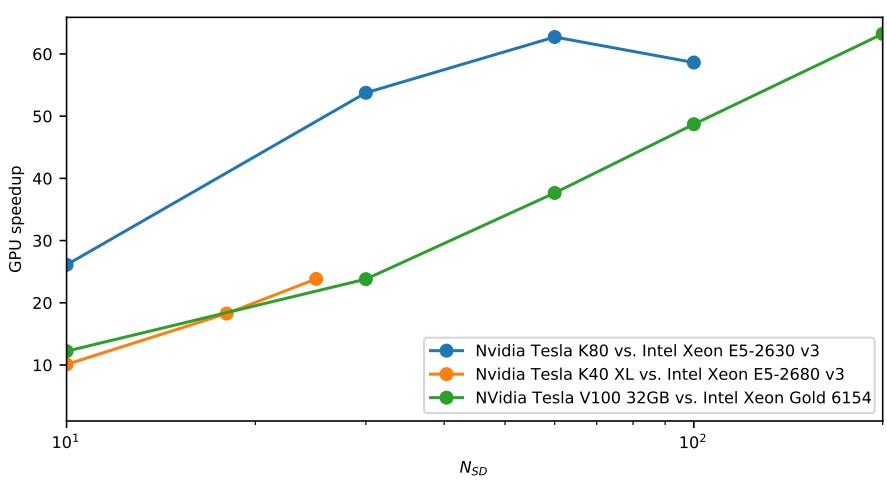

**Figure 3.** Number of CPUs needed to replace single GPU to obtain the same wall time, assuming that wall time scales perfectly with added CPUs. Based on single-node simulations.



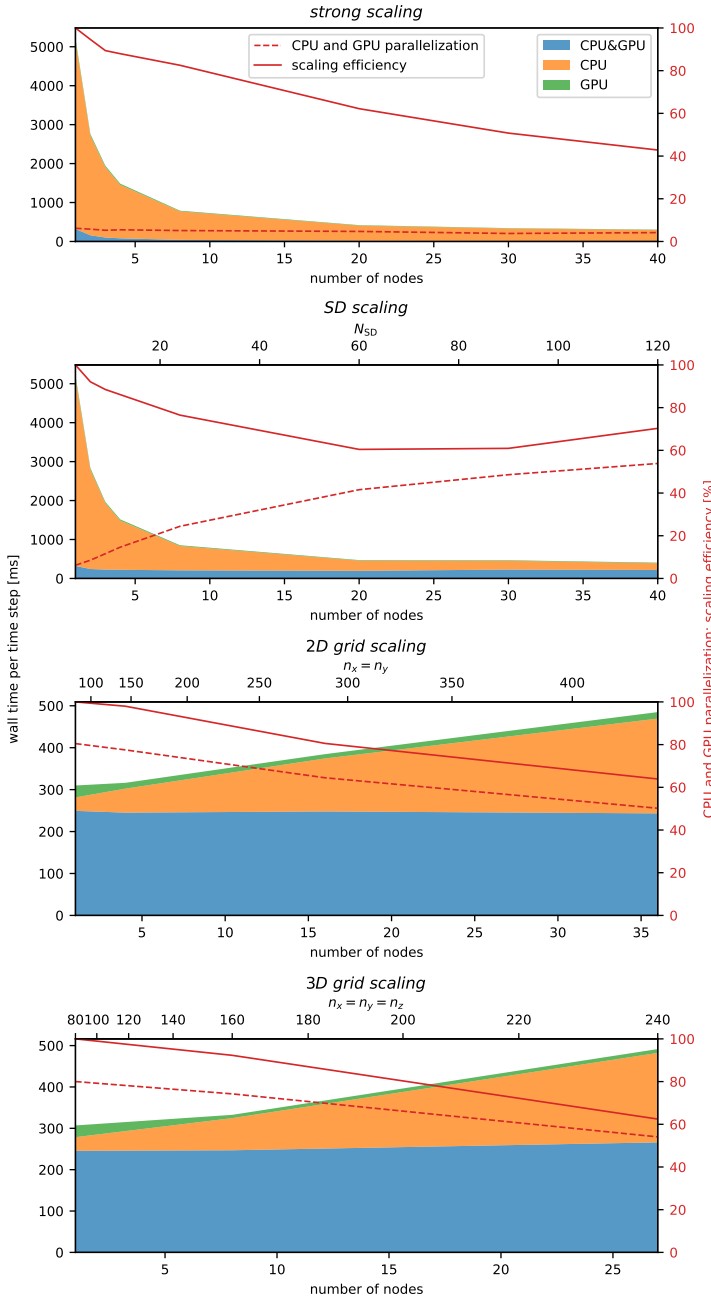

**Figure 4.** Multi-node UWLCM-SDM performance. Wall time per model time step averaged over 100 time steps. Results of LES of a raising thermal done on the *Prometheus* cluster for different scaling scenarios. Execution time is divided into time spent on CPU computations, GPU computations and parallel CPU and GPU computations. The dashed red line is the percentage of time spent on parallel CPU and GPU computations. The solid red line shows scaling efficiency, defined as the wall time assuming perfect scaling divided by the actual wall time.



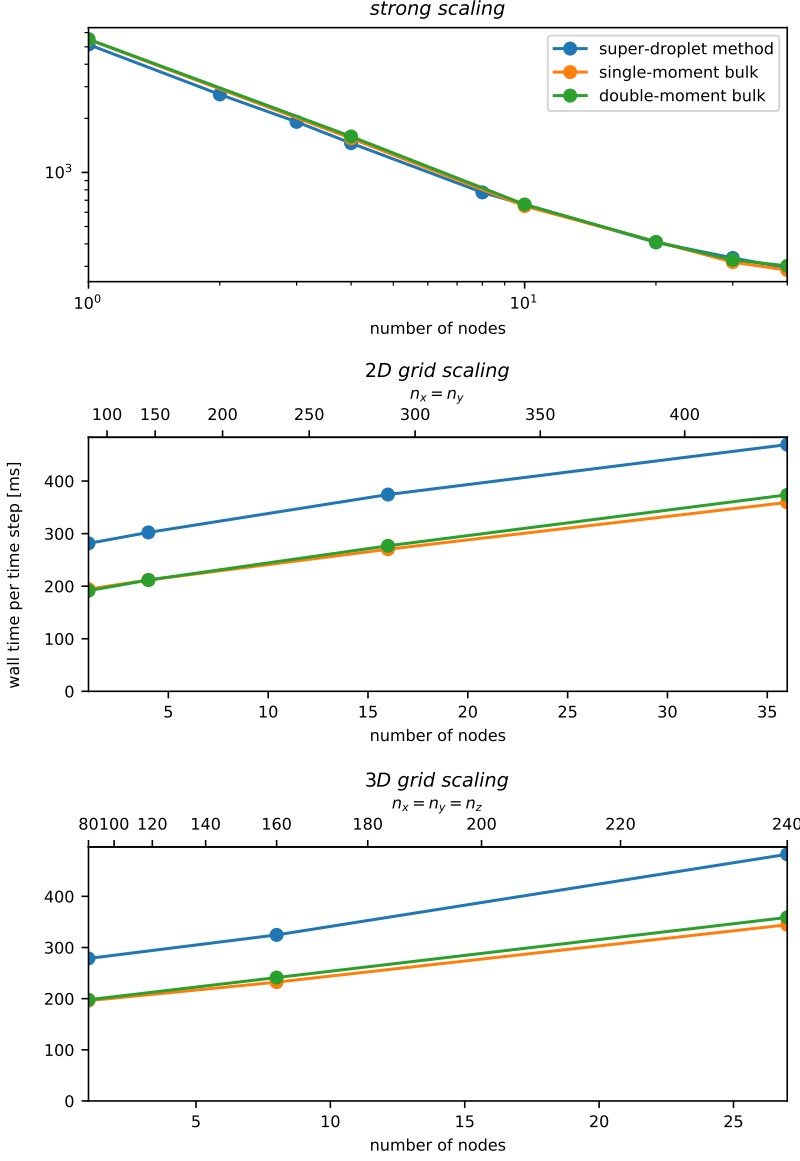

**Figure 5.** Multi-node model performance for different microphysics schemes. Wall time per model time step averaged over 100 time steps. Results of LES of a raising thermal done on the *Prometheus* cluster for different scaling scenarios.



**Table 1.** List of hardware of servers used.

| Name | CPU | CPU cores | GPU | GPU memory |
|---|---|---|---|---|
| *Rysy* | 2x Xeon Gold 6154 @ 2.50 GHz | 2x18 | 4x Tesla V100 | 4x32GB |
| *a02* | 2x Xeon E5-2630 v3 @ 2.40GHz | 2x8 | 2x Tesla K80 | 2x24GB |
| *Prometheus*[a] | 2x Xeon E5-2680 v3 @ 2.50 GHz | 2x12 | 2x Tesla K40 XL | 2x12GB |

[a] The cluster has 72 such nodes.





**Table 2.** List of software of servers used.

| Name | CUDA | gcc | Boost | HDF5 | Thrust | blitz++ |
|---|---|---|---|---|---|---|
| *Rysy*[a] | 11.0 | 9.3.0 | 1.71.0 | 1.10.4 | 1.9.5-1 | 1.0.2 |
| *a02* | 10.1 | 4.8.5 | 1.60.0 | 1.8.12 | 1.9.7 | 0.10 |
| *Prometheus* | 11.2 | 9.3.0 | 1.75.0 | 1.10.7 | 1.10.0 | 1.0.2 |

[a] software from the Singularity containter distributed with UWLCM.





**Table 3.** Details of multi-node scaling tests. $n_x$, $n_y$ and $n_z$ is the total number of Eulerian grid cells in the respective direction. $N_{\mathrm{SD}}$ is the initial number of super-droplets per Eulerian grid cell. Workload per CPU is estimated assuming that it is proportional to the number of grid cells per CPU only. Workload per GPU is estimated assuming that it is proportional to the number of super-droplets per GPU only. MPI transfers, data transfers between CPU and GPU memories, and GPU handling of information about Eulerian cells are not included in these workload estimates. Data transfer sizes are for copies between different MPI tasks, but do not include copies between CPU and GPU memories of the same task. Data transfer sizes are estimated assuming that time step length and air flow velocities do not change with grid size. $t^1$ is the wall time on a single node. $t^2_{\mathrm{GPU}}$ is the wall time of GPU and CPU&GPU calculations in a simulation on two nodes.

| | *strong scaling* | *SD scaling* | *2D grid scaling* | *3D grid scaling* |
|---|---|---|---|---|
| $n_x$ | 240 | 240 | $\sqrt{N_{\mathrm{nodes}}} \times 72$ | $\sqrt[3]{N_{\mathrm{nodes}}} \times 80$ |
| $n_y$ | 240 | 240 | $\sqrt{N_{\mathrm{nodes}}} \times 72$ | $\sqrt[3]{N_{\mathrm{nodes}}} \times 80$ |
| $n_z$ | 240 | 240 | 100 | $\sqrt[3]{N_{\mathrm{nodes}}} \times 80$ |
| $N_{\mathrm{SD}}$ | 3 | $N_{\mathrm{nodes}} \times 3$ | 100 | 100 |
| Eulerian cells in domain $[10^3]$ | 13824 | 13824 | $N_{\mathrm{nodes}} \times 518.4$ | $N_{\mathrm{nodes}} \times 512$ |
| superdroplets in domain $[10^6]$ | 41.472 | $N_{\mathrm{nodes}} \times 41.472$ | $N_{\mathrm{nodes}} \times 51.84$ | $N_{\mathrm{nodes}} \times 51.2$ |
| workload per CPU | $\propto 1/N_{\mathrm{nodes}}$ | $\propto 1/N_{\mathrm{nodes}}$ | const. | const. |
| workload per GPU | $\propto 1/N_{\mathrm{nodes}}$ | const. | const. | const. |
| data transfer size per CPU | const. | const. | $\propto \sqrt{N_{\mathrm{nodes}}}$ | $\propto N_{\mathrm{nodes}}^{2/3}$ |
| data transfer size per GPU | const. | $\propto N_{\mathrm{nodes}}$ | $\propto N_{\mathrm{nodes}}$ [a] | $\propto N_{\mathrm{nodes}}$ [a] |
| time assuming perfect scaling | $t^1/N_{\mathrm{nodes}}$ | $\max(t^1/N_{\mathrm{nodes}}, t^2_{\mathrm{GPU}})$ [b] | $t^1$ | $t^1$ |

[a] Assuming that grid scaling is used to refine the resolution, as done in this paper.
   If it is done to increase the domain, data transfer size per GPU scales as the one per CPU.

[b] GPU time from two nodes simulation is taken as reference, because it is ca. 25% lower
   than on a single node. A plausible explanation for this is that, although the number of SD
   per GPU does not depend on the number of nodes, GPUs also store information about conditions
   in grid cells, and the amount of grid cells per GPU decreases as more nodes are used.
   For more than 2 nodes, GPU calculation time is approximately the same as for 2 nodes.





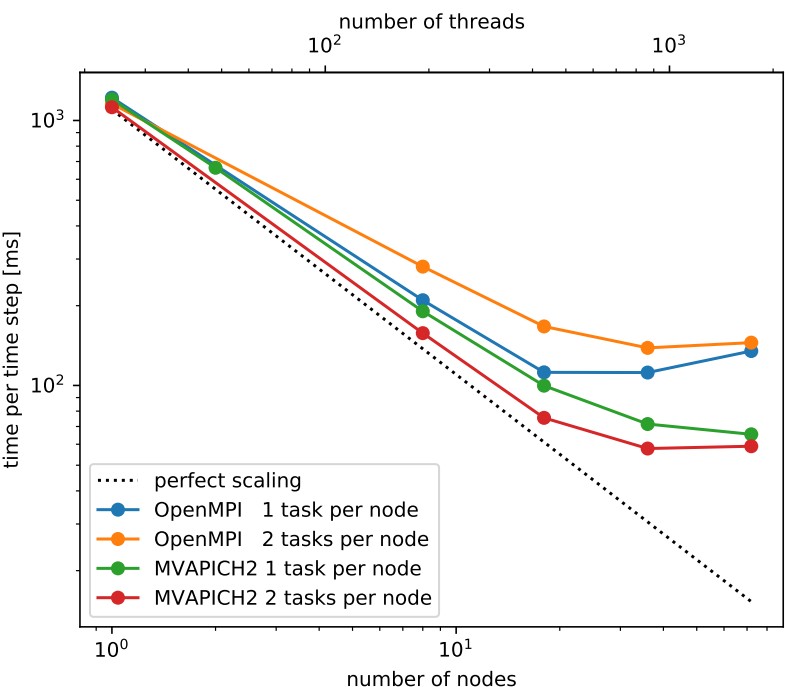

**Figure A1.** Strong scaling test of the libmpdata++ library. Wall time per time step of a dry planetary boundary layer simulation. The dotted black line shows perfect scaling.



**Figure A2.** UML sequence diagram showing the order of operations in the UWLCM 2.0 model. Right-hand-side terms are divided into condensational, non-condensational and subgrid-scale parts, $R = R_c + R_n + R_{\mathrm{SGS}}$. Other notation follows Dziekan et al. (2019).