# Peer review of "University of Warsaw Lagrangian Cloud Model (UWLCM) 2.0: Adaptation of a mixed Eulerian-Lagrangian numerical model for heterogeneous computing clusters"

_Geoscientific Model Development, 2021_

## Author Comment (AC1)

**We thank the reviewer for the comments. Before we answer the comments, we need to point out that we found an error in model runs on the Prometheus cluster. The error caused collision-coalescence to be not modeled and made condensation time step to be 0.5 s. In typical cloud simulations collision-coalescence is modeled and condensation time step is around 0.1 s. The error significantly reduced time complexity of microphysics computations. The error has been fixed and simulations were repeated. There are important differences between the faulty and corrected Prometheus results. Discussion of results, summary and abstract have been reworked to account for these differences.**

**Answers to the reviewer comments are listed below.**

General Comments:

The paper describes a port to GPU for the University of Warsaw Lagrangian Cloud Model (UWLCM). The authors have identified a sensible separation, handling Eulerian terms (bulk flow) with CPU computing and offloading the Lagrangian terms (particle) to the GPU. Alongside the effort involved in rewriting the code for GPU, the careful splitting of terms and balancing the workload between CPU and GPU are this paper's main original contribution.

Section 1 gives a good overview of cloud modelling  certainly suitable for those unfamiliar with the topic. Combined with section 2 this gives a brief but sufficient overview of some cloud modelling literature and UWLCM. Section 3 gives a good description of the authors contributions in terms of code modification strategy.The breakdown in section 4 describing simulation hardware and the performance results is good and the level of detail is suitable for most of the analysis.

The results look very promising, but would benefit from some significant changes to the presentation and corresponding modification in the discussion in the text. (See specific comments). Reproducibility is also considered, the authors' code is available, the system stack is fully specified as well as hardware used.

Specific Comments:

There are some key areas where changes need to be made:

1. One of the key contributions of this paper is the author's careful handling of balance between CPU and GPU workload. Specifically, trying to simultaneously utilise both the CPU and GPU as much as possible. However, the treatment of this in both the text and figures needs further clarification.

In the abstract, this was unclear and this reviewer thought it was a mistake, a rewording here would clarify the term. The text should also have a definition of what the authors intended with the term "amount of parallelisation".

This lack of clear definition is compounded by Figures 2 and 4. These figures should be modified to present the information more clearly.

As I understand the total wall time per timestep (t_tot) comprises of:

t_tot = t_both + t_CPU + t_GPU

Where

 - t_both is the data presented in blue (CPU + GPU) the time that the authors are aiming to maximise to increase the "amount of parallelisation".

 - t_CPU is time where only the CPU is performing work and the GPU is idle

 - t_GPU is time where only the GPU is performing work and the CPU is idle

**That is correct. We modified the abstract so that it does not include the term "amount of parallelisation". A new subsection "Performance metrics" has been added to the manuscript. In it we define t_tot, t_CPU, t_GPU, t_CPU&GPU and what we mean by "parallelization". In Figs. 2 and 4 ambiguous terms have been replaced with t_tot, t_CPU, t_GPU or t_CPU&GPU.**

My suggestion for improving the figure is having these quantities stacked, rather than overlapping, either as a shaded chart or preferably a stacked bar chart. This would more clearly present the t_tot quantity of interest, which is currently hidden. The structure of the plot should also be outlined in the text.

If any of my assumptions here are incorrect, there should be clarification in the text as to what is actually being plotted.

**Values of t_CPU, t_GPU and t_both are stacked in Figures 2 and 4. This has been clarified in figure captions. We use a stacked shaded chart instead of a stacked bar chart, because values on the horizontal axis are not uniformly distributed and the shaded chart is more readable.**

I am not opposed to the auxiliary plot in red, but in Figure 2 the y-limits should be fixed as in Figure 4 so comparisons between the plots can be made.

**The y-limits have been fixed as suggested.**

2. In two places in the text, power and energy are mentioned but not discussed. Either, these comments should be removed as there is no backup or discussion of the assertions. Or, power information for both CPUs and GPUs needs to be presented in table 1 and at least an estimation of energy/power usage needs to be presented and a comparison between CPU vs CPU+GPU made.

**Information about thermal desing power (TDP) of CPUs and GPUs has been added to table 1. Estimated energy used in CPU and CPU+GPU is now plotted in Fig. 3.**

3. The discussion from lines 150-156 along with Figure 3 doesn't really make sense. CPUs and GPUs cannot be compared in such a way, and the assertion that a certain number of CPU cores are in any way equivalent to a GPU is misleading, since the basis of comparison is wall time only. This doesn't take into account aspects like power consumption or accuracy, for instance. From the discussion, the quantity of interest here is total time (per timestep perhaps), this quantity should be plotted for both CPU only and CPU+GPU case so readers can compare. Similar figures could be

used to compare the energy/power usage for CPU vs CPU+GPU and the accuracy, should the authors desire to draw a comparison.

**The CPU and GPU comparison has been changed. In Fig. 3, wall time and energy use per time step are now compared, instead of the hypothetical number of CPUs that would replace GPUs. The discussion has been changed accrodingly.**

Whilst the changes to the plots may be significant, I believe that they can be made without performing additional simulations, just by changing plotting scripts. However, I believe that the changes should be further reviewed so I have suggested major, rather than minor revisions.

More minor corrections, line by line notes:

L4: (Abstract) Amount of parallelization, needs clarification. *See 1 above

**Corrected, see answers above.**

L10: Moore's Law hasn't really ended, Dennard scaling has, I can't take too much issue here though, as the cited paper backs up your point.

**This has been replaced with:**

**"As CPU clock frequencies no longer stably increase over time and the cost per transistor increases, (...)"**

L89: Some expected speeds should be given for expected PCI express and interconnect speeds, for comparison (could be added to table)

**PCI-E and interconnect speeds are now given in table 1.**

L90-91: Clarification, what technology is allowing GPU-GPU inter-node communication? This should be mentioned.

**We added:**

**"Intra-node communication between GPUs controlled by a single process makes use of the NVIDIA GPUDirect Peer to Peer technology, which allows direct transfers between memories of different devices. Communication between GPUs controlled by different processes is handled by the MPI implementation. If the MPI implementation uses the NVIDIA GPUDirect Remote Direct Memory Access technology, inter-node GPU-to-GPU transfers go directly from device memory to the interconnect, without host memory buffers."**

Table 1: Please add memory bandwidth figures for both main memory (RAM) and GPU memory as well as theoretical peak flops for CPUs and GPUs.

**Done**

L156: Power usage is mentioned here with no prior discussion or further explanation. Either remove comment about power, or include wattage for CPU + GPU in table 1 and discuss this fully. *See 2 above

**See answer to comment 2.**

L217: Energy here is again a throw away comment. *See 2 above

**See answer to comment 2.**

L223: "A simulation with 20 million grid cells and 2 billion particles" I like the discussion of the number of DOFs here, the earlier text and possibly also figures would also benefit from some concrete discussion of the problem size. (I'm aware of table 3, but would be useful to have some discussion in text)

**We added:**
**"Depending on the scenario, number of Eulerian grid cells is between 0.5 and 18.5 million, and number of Lagrangian particles is between 40 million and 18.5 billion."**
**to the discussion of testing scenarios in section 4.5.**

Figure 5: Would benefit from a line indicating the ideal scaling in each case.

**Such line has been added.**

Technical Corrections:

**All suggested technical corrections were made.**

I have ignored any British English vs American English discrepancies.

What follows are my interpretations/suggestions for typos and grammar:

Parallell -> Parallel (throughout)

L18-19: what is known as cloud parameterization -> which is known as cloud parameterization

... and throughout "what" -> "which" L22 for example (what requires) L27, L65, L183

L31 gained in -> gained

L37 Main goals -> The main goals

L71 parallely to -> in parallel with

L79 Strategy -> The strategy

Table 2: List of software of servers used -> List of software on servers used. (I think this is what was meant)

L129 ran -> run ... and throughout L134,L159,L178

L209: parallelly done -> done in parallel

**Additional changes**

**We fixed an error in the link to the UWLCM code.**

---

## Author Comment (AC2)

**We thank the reviewer for the comments. Before we answer the comments, we need to point out that we found an error in model runs on the Prometheus cluster. The error caused collision-coalescence to be not modeled and made condensation time step to be 0.5 s. In typical cloud simulations collision-coalescence is modeled and condensation time step is around 0.1 s. The error significantly reduced time complexity of microphysics computations. The error has been fixed and simulations were repeated. There are important differences between the faulty and corrected Prometheus results. Discussion of results, summary and abstract have been reworked to account for these differences.**

**Answers to the reviewer comments are found below.**

The authors have presented results from experiments performed to evaluate the performance of a Eulerian-Langragian numerical model adapted to heterogeneous computer systems. Since the original model is not new, I find the description of scientific and numerical methods used in the model satisfying and well referenced.

A consideration is made for the fact that the authors used hardwares and softwares which are readily available to them to conduct the experiments whose results are presented. I however find some of their assumptions and technical arguments used to arrive at their conclusions misleading and confusing. For example, in line 155 the authors write "We conclude that GPUs provide substantial benefits in equipment cost and power usage" while there are no results shown in the manuscript to support this conclusion. Until the authors clearly show data on the cost and power consumption of the GPUs and CPUs used in the experiments, the statement should be removed.

**Argument about equipment cost has been removed. Power consumption argument is now supported by a plot of energy used by a CPU-only vs CPU+GPU simulation (Fig. 4).**

Furthermore, I find the following terms used ambiguously throughout the manuscript, parallelization, system memory, server, and complexity. The terms are in some cases used in a non-standard way making it difficult for the reader to interpret the results presented. My comments in this regard are as follows:

- **Parallelization**: the authors have used the term "parallelisation" multiple times in the manuscript, e.g. in lines 4, 187, 195, 212, 222 as well as figures 2 and 4. The UML sequence diagram in figure A2 however shows concurrency which is not necessarily a parallelization. The fact that the original model also contain some aspect s of parallel programming may cause confusion to readers when the authors refer to the sequence shown in figure 2 as a "CPU and GPU parallelization". Perhaps "Concurrent CPU-GPU operations" will fit well.

   **We use "parallelization" to describe the percentage of time when CPU and GPU perform tasks at the same time, what we believe is the correct use of the term. This is described in a new subsection "Performance metrics". Parallell computations are done during some fraction of the entire time step shown in the UML diagram. Timer measuring GPU computations is started when submitting a task to GPU and stopped when the task finishes.**

- **System memory:** the authors have used the term "system memory" in a way that leaves the reader with no clear picture of the memory model applied. In lines, 88, 192, and 211 they

use "system memory" to refer to the memory only accessed by CPU. Then in line 88 to 89 the authors write "Since the CPU and GPU data attributed to a task are colocated in the modeled space, all CPU-to-GPU and GPU-to-CPU communications happen via PCI-Express…", and in line 135 they write "….all Eulerian fields are stored in shared memory". These statements considered together with the previously mentioned lines and put into context with "GPU memory" mentioned in lines 71, 84, 134, 162, 173, 181, and table 1, is confusing. To avoid this confusion the authors should adopt the standard heterogeneous memory model where "host memory" and "device memory" refers to CPU and GPU memory spaces respectively.

**As suggested, we adopted the "host memory" and "device memory" terminology.**

- **Server**: the authors have used the term "server" to refer to a single computational unit defined by a memory configuration in lines 37, 129, 135, 144, 153, 211, 217, figure 2, and tables 1 and 2. At the same time the authors use the same argument to introduce "single-node" in section 4.3 and "multi-node" in section 4.4. It will be clear for the reader if the authors consistently used single-node and multi-node systems as the standard definitions for shared and distributed memory units as opposed to using the term "server".

  **"Server" has been replaced with "single-node system" or "multi-node system".**

- **Complexity**: the authors have used "complexity" to loosely refer to time complexity in lines 139 and 140, but they seem to acknowledge the role of space complexity in lines 144 and 145. The two should be clearly separated and if possible shown using well defined or derived mathematical functions or presented graphically.

  **Space and time complexity are now separated and a citation about complexity of Lagrangian microphysics has been added.**

Additionally, the following lines should be corrected, rewritten or removed due to ambiguity and misspellings:

88: The term "system memory" is ambiguous here considering the standard heterogeneous computing memory model. "Host memory will be appropriate. Table 1 should also show the amount of memory available to the hosts in the systems described. See above the comment on system memory.

**"System memory" has been replaced with "host memory". Host memory is now listed in table 1.**

92 - 93: "An MPI task will typically control more than one CPU thread, because usually cluster nodes have more CPU cores than GPUs" This statement is misleading if you consider the current and future GPU clusters.

**The sentence has been removed.**

94 - 95: The statement should start with "The maximum number..." and end with "... in the x direction.

**Fixed.**

123 - 124: The performance need some comparison to be labeled "lower".

**"Lower performance" has been replaced with "greater simulation wall time". Simulation wall time is shown in Fig. A1.**

126 - 127: What threads do the authors refer to here? GPU or CPU threads? This should be clarified.

**It is now clarified that these are CPU threads.**

129: Perhaps a "single-node system" instead a "server" is more fitting. See the comment on "server" above.

**Fixed, see "server" comment.**

132: The authors write "…grid is 128x128x128". Are these grid cells? Proper definition is necessary.

**It is now clarified that these are grid cells.**

134 - 135: It is not clear which shared memory the authors refer to here, whether in the host or device. Also consider the comment on "server" above.

**The sentence has been removed.**

136:  Figure reference is missing and "where" should be replaced with "were".

**The sentence has been removed.**

136 - 137: The authors should clarify whether there is any overhead in the timing function(s). If there is how does it affect the numbers shown?

**There is no difference in simulation wall time with and without the timing functions. This is now written in section 4.3.**

136 - 138: The authors should show the analyses which were done with ARM MAP and VTune, and indicate how the numbers compare to the results from the UWLCM function. Otherwise the sentence should be removed.

**MAP and VTune were used to find which part of the code is responsible for decrease in scaling efficiency. The result was that collective MPI calls in the pressure solver scale worst. Unfortunately, these profilers are no longer available on the Prometheus cluster, so we were unable to repeat the analysis for new, corrected runs. Therefore the sentence has been removed.**

137: Replace "where" with "were".

**Corrected**

138: Replace "vTune" with "Vtune"

**Corrected**

139 - 141: Usage of the term "complexity" is ambiguous here, the authors should clarify whether they are talking about time or space complexity. See the comment on complexity above.

**Clarified as described in the answer to the comment on complexity.**

140 - 141: The authors should support the statement on complexity of GPU computations with a well defined or derived mathematical relation or graphically. See comment on complexity above.

**Clarified as described in the answer to the comment on complexity.**

147 - 148: It is not clear what the authors refer to as parallelization of CPU and GPU computations (also in line 187). Should be rewritten for clarity. See comment on parallelization above.

**Parallelization measure is now defined in a new section. See answer to the comment above.**

150: The sentence should read "how much speedup is achieved by employing GPU resources".

**Corrected**

153: The authors should reconsider the usage of the term "server". See the comment on server above.

**Clarified as described in the answer to the comment above.**

154: There is no clear mathematical relationship between what the authors have defined as the "GPU speedup" and $N_{SD}$ making the interpretation of figure 3 difficult. Should be revised accordingly.

**The sentence has been rewritten as fig. 4 now shows wall time and energy instead of the speedup.**

155: This conclusion is not supported by any result. Without data on the cost and power consumption of the used hardware, the sentence should be removed.

**Power consumption is now plotted. Comment about equipment cost has been removed.**

161 - 171: Reference to table 3 should come earlier in line 160 to make the description of the mentioned configurations more under stable.

**Done**

2019 - 2020: 'higher factor' should be substantiated with a comparison.

**We added:**

**"For example, in a weak scaling scenario (3D grid scaling) $t^{tot}_{CPU}$ is approximately three times larger on 27 nodes than on one node, while $t^{tot}_{GPU}$ is increased by only around 7% (fig. 5).**

223: The authors write "A simulation with 20 million grid cells and 2 billion particles can be done in real time" without any evidence. The sentence should be removed unless it is substantiated with clear evidence or references.

**This claim was based on the old fig. 5 that depicted faulty simulations, in which there was less GPU workload than should be. This sentence has been removed from the reworked summary.**

227 - 229: This paragraph should be revised to include clear meaning of parallelization. See the comment on parallelization above.

**In the revised paragraph we write about the amount of time during which CPUs and GPUs compute at the same time in place of parallelization.**

Appendix B: The authors use "processes" and "tasks" ambiguously. The paragraph should be revised to remove the ambiguity.

**"Task" has been replaced with "process".**

Appendix C: Should be removed and the paragraph included in section 4.

**Appendix C has been removed. The paragraph is now included in section 2 "Model description". Section 4 "Performance tests" does not seem to be the appropriate place to discuss model details.**

Table 3:

- Definition of $N_{nodes}$ is missing.

  **Definition ($N_{nodes}$ is the number of nodes) is now given in table caption.**

- Is there any relationship between $n_x$, $n_y$, $n_z$ and "Eulerian cells in domain [$10^3$]; and $N_{SD}$ and "super droplets in domain [$10^6$]? If there is, it should be described otherwise the table is confusing.

- **Number of Eulerian grid cells in the domain is equal to $n_x \times n_y \times n_z$ . Number of superdroplets in the domain is equal to $n_x \times n_y \times n_z \times N_{SD}$. This is now clarified in table caption.**

**Additional changes**

**We fixed an error in the link to the UWLCM code.**